# GABA$_B$R silencing of nerve terminals

Daniel C Cook[1], Timothy A Ryan[1,2]*

[1]Department of Anesthesiology, Weill Cornell Medical College, New York, United States; [2]Department of Biochemistry, Weill Cornell Medical College, New York, United States

**Abstract** Control of neurotransmission efficacy is central to theories of how the brain computes and stores information. Presynaptic G-protein coupled receptors (GPCRs) are critical in this problem as they locally influence synaptic strength and can operate on a wide range of time scales. Among the mechanisms by which GPCRs impact neurotransmission is by inhibiting voltage-gated calcium ($Ca^{2+}$) influx in the active zone. Here, using quantitative analysis of both single bouton $Ca^{2+}$ influx and exocytosis, we uncovered an unexpected non-linear relationship between the magnitude of action potential driven $Ca^{2+}$ influx and the concentration of external $Ca^{2+}$ ($[Ca^{2+}]_e$). We find that this unexpected relationship is leveraged by GPCR signaling when operating at the nominal physiological set point for $[Ca^{2+}]_e$, 1.2 mM, to achieve complete silencing of nerve terminals. These data imply that the information throughput in neural circuits can be readily modulated in an all-or-none fashion at the single synapse level when operating at the physiological set point.

## Editor's evaluation

In this important study Cook and Ryan find that at physiological temperatures, small changes in extracellular calcium concentrations shift individual presynaptic terminals between active and inactive states. This discovery is extended to the action of inhibitory synaptic modulation via GABA, emphasizing potentially broad relevance for our understanding of how synaptic transmission is stabilized and modulated throughout the brain. The evidence for the observed phenomenon is compelling, given the use of multiple functional reporters and rigorous analyses.

*For correspondence:
taryan@med.cornell.edu

Competing interest: The authors declare that no competing interests exist.

## Introduction

GPCRs serve a key control function in the brain, modifying the efficacy or time course of neurotransmission through pre- and postsynaptic mechanisms (*Nadim and Bucher, 2014*; *Lovinger et al., 2022*). Ligands of GPCRs are often secreted near synapses and exert their effects locally (*van den Pol, 2012*; *van Westen et al., 2021*; *Ding et al., 2019*). They can thereby modify region-specific synaptic properties and change the weighting of targeted synapses, in turn altering local computation. A canonical mode of presynaptic modulation occurs via the binding of voltage-gated calcium channels (VGCCs) by G-protein βγ subunits after they dissociate from the trimeric G-protein complex following GPCR activation (*Herlitze et al., 1996*). The impact of GPCR-mediated inhibition of VGCCs on neurotransmission (*Kreitzer and Regehr, 2000*; *Yamada et al., 1999*) is amplified by the exquisite sensitivity of exocytosis to the magnitude of $Ca^{2+}$ influx, which underlies the ability to rapidly change exocytotic rates at sites of neurotransmitter release. This sensitivity is driven by the cooperative binding of multiple $Ca^{2+}$ ions to one or more $Ca^{2+}$ sensors that form core elements of the exocytic machinery (*Heidelberger et al., 1994*; *Schneggenburger and Neher, 2000*; *Bollmann et al., 2000*; *Ariel and Ryan, 2010*). Modifying the open probability of VGCCs, therefore, serves as a potent means of presynaptic control, as small changes in $Ca^{2+}$ influx can lead to large changes in neurotransmitter release.

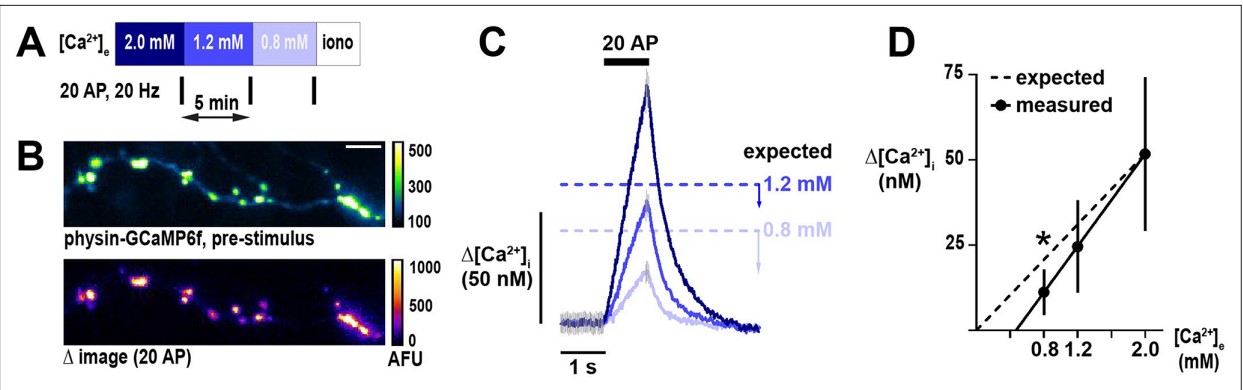

**Figure 1.** Unexpected sub-proportionality in dependence of presynaptic calcium ($Ca^{2+}$) influx on $[Ca^{2+}]_e$. (**A**) Schematic of the experimental protocol. Neurons are stimulated with a brief action potential (AP) train (20 AP in 1 s) at three different $[Ca^{2+}]_e$ centered on the physiologic set point of 1.2 mM. Ionomycin, a $Ca^{2+}$ ionophore, is administered at the end of the experiment to convert fluorescence into absolute $[Ca^{2+}]_i$. (**B**) An axon from a neuron expressing GCaMP6f localized to nerve terminals by synaptophysin (physin-GCaMP) at rest (top) and a difference image showing the response to 20 AP (bottom). The difference image is the mean peak (five frames) subtracted by the mean baseline (49 frames) in $[Ca^{2+}]_e$ 2 mM. For display, a representative subset of terminals was highlighted. Scale bar is 10 μm. (**C**) Traces of responses to 20 AP from the neuron shown in B, with $[Ca^{2+}]_e$ color-coded as in A. Expected $\Delta[Ca^{2+}]_i$ at 0.8 and 1.2 mM is represented by dashed lines and calculated as a change in influx relative to 2.0 mM proportional to the ratios of $[Ca^{2+}]_e$. Arrow bars show the difference between expected and measured $\Delta[Ca^{2+}]_i$. Traces are mean, with error (SEM) represented by gray lines in only pre-stimulus and peak frames for clarity (n=211 nerve terminals). (**D**) Summary of $\Delta[Ca^{2+}]_i$ as a function of $[Ca^{2+}]_e$ (mean ± 95%CI, n=9 neurons). The x-axis intercept for measured changes in $\Delta[Ca^{2+}]_i$ is 0.47 mM. *p<0.05, one-sample *t*-test compared to the expected value.

The online version of this article includes the following figure supplement(s) for figure 1:

**Figure supplement 1.** $Ca^{2+}$ entry into nerve terminals is nearly completely abolished at a non-zero threshold.

The impact of $Ca^{2+}$ on exocytosis depends on local changes in $Ca^{2+}$ concentration near the active zone, which in turn depend on the external $Ca^{2+}$ concentration bathing the nerve terminal (*Dittman and Ryan, 2019*). Thus, the steepness of the relationship between exocytosis rates and $Ca^{2+}$ influx depends crucially on resting $[Ca^{2+}]_e$. In vivo, plasma $[Ca^{2+}]$ is strictly regulated, and the CNS is very sensitive to plasma $[Ca^{2+}]$ perturbations, with seizures precipitated by hypocalcemia and psychiatric symptoms progressing to lethargy or coma with worsening hypercalcemia (*Riggs, 2002*). Feedback loops controlled by $Ca^{2+}$ sensors of the parathyroid gland act to maintain plasma $[Ca^{2+}]$ at a set point of 1.2 mM in most mammals (*Hannan et al., 2019*). Here, we used quantitative optical tools to investigate the sensitivity of exocytosis rates at the single-synapse level when operating at the physiological set point for $[Ca^{2+}]_e$. In a narrow range of $[Ca^{2+}]_e$ near the physiologic concentration, our experiments revealed an unexpected sub-proportionality of $Ca^{2+}$ influx relative to $[Ca^{2+}]_e$, indicating that a minimal $Ca^{2+}$ entry is needed to sustain robust $Ca^{2+}$ influx during AP firing. We show that three different approaches to modestly decrease $Ca^{2+}$ influx, (1) blocking a fraction of $Ca^{2+}$ channels, (2) lowering $[Ca^{2+}]_e$, or (3) application of a GPCR agonist, baclofen, known to reduce $Ca^{2+}$ influx lead to complete silencing of a subset of nerve terminals. Thus, in this operating regime, decreasing presynaptic $Ca^{2+}$ current may act as a digital switch to eliminate single bouton synaptic throughput.

## Results

### $Ca^{2+}$ influx into nerve terminals is not 1:1 proportional to $[Ca^{2+}]_e$ near the physiologic set point

In order to examine how $Ca^{2+}$ influx driven by action potentials (APs) at nerve terminals is impacted by changes in $[Ca^{2+}]_e$, we used the genetically-encoded $Ca^{2+}$ indicator physin-GCaMP6f transfected sparsely into primary cultures of dissociated rat hippocampal neurons (*Figure 1*). We measured the changes in the GCaMP6f signal in response to a 20 AP burst stimulus (20 Hz) at 0.8, 1.2, and 2.0 mM $[Ca^{2+}]_e$ (*Figure 1A and B*). Signals were corrected for the local abundance of the protein by determining the GCaMP6f fluorescence under saturating $Ca^{2+}$ binding (determined following ionomycin application) and converted to absolute intracellular $Ca^{2+}$ concentrations ($[Ca^{2+}]_i$, see methods). Previous studies investigating the dependency of $\Delta[Ca^{2+}]_i$ to $[Ca^{2+}]_e$ in nerve terminals of the CNS

found the relationship obeyed a simple saturable pore model with a $K_d$ of $[Ca^{2+}]_e$ ~2.5 mM (*Ariel and Ryan, 2010*; *Schneggenburger et al., 1999*). This model predicts that changes in $\Delta[Ca^{2+}]_i$ will exhibit 1-to-1 proportionality to $[Ca^{2+}]_e$ at concentrations well below $K_d$, which is substantially higher than the physiologic set point ($[Ca^{2+}]_e$ = 1.2 mM). One important confound, however, is that previous measurements were not carried out at physiological temperatures. Here, where the temperature was set to 37 °C, we observed an unexpected sub-proportional relationship of $\Delta[Ca^{2+}]_i$ to $[Ca^{2+}]_e$ as the concentration was lowered below 2.0 mM (*Figure 1B and C*). The magnitude of difference between the expected and experimentally measured $\Delta[Ca^{2+}]_i$ was greater at $[Ca^{2+}]_e$ 0.8 mM compared to 1.2 mM, highlighting that this discrepancy manifests most prominently below the set point (*Figure 1C*). A linear extrapolation from the mean $\Delta[Ca^{2+}]_i$ measurements relative to $[Ca^{2+}]_e$ predicts the cessation of $Ca^{2+}$ entry into nerve terminals at $[Ca^{2+}]_e$ of 0.47 mM (*Figure 1C*). Experiments carried out using $[Ca^{2+}]_e$ = 0.4 mM confirm that there is almost no measurable residual $Ca^{2+}$ influx under these conditions (estimated $\Delta[Ca^{2+}]_i$~1.6 nM; *Figure 1—figure supplement 1*), far below the value expected (10.4 nM) if $Ca^{2+}$ influx obeyed a simple linear proportionality.

## Modest changes of $[Ca^{2+}]_e$ near the physiologic set point controls the proportion of silent nerve terminals

The above measurements represent ensemble averages of all terminals with a measurable signal following the saturation of physin-GCaMP. To further probe the apparent non-zero threshold of $[Ca^{2+}]_e$ that enables $Ca^{2+}$ entry, we assessed physin-GCaMP responses at individual nerve terminals with varying $[Ca^{2+}]_e$ (*Figure 2*). Surprisingly, we found that a subset of terminals exhibited an all-or-none response to lowering $[Ca^{2+}]_e$ below the physiologic set point such that $\Delta[Ca^{2+}]_i$ is selectively abolished (*Figure 2B*). To rigorously quantify this behavior, we employed a modest cutoff of one standard deviation of the pre-stimulus noise above the mean fluorescence to separate responding terminals from those without measurable synaptic activity, termed silent (*Moulder et al., 2008*; *Altrock et al., 2003*; *Kim and Ryan, 2010*). This definition of presynaptic silencing has been used previously in neurons expressing GCaMP3, a less sensitive indicator than GCaMP6f, and compared to the low-affinity, synthetic $Ca^{2+}$ dye Magnesium Green with an excellent agreement, indicating that GCaMP sensors can reliably distinguish responding and silent terminals despite their non-linear $Ca^{2+}$ dependence (*Kim and Ryan, 2013*). To evaluate the effectiveness of our thresholding, we compared cumulative frequency curves of $\Delta F$ for terminals separated into silent and responding populations (*Figure 2—figure supplement 1*). Responding terminals exhibit a clear shift towards higher $\Delta F$ with increasing $[Ca^{2+}]_e$, as expected, but the distribution of silent terminals was invariant with respect to $[Ca^{2+}]_e$. This analysis demonstrates that silent terminals are appropriately classified as incapable of generating detectable $Ca^{2+}$ influx that would otherwise scale with $[Ca^{2+}]_e$. We observed that silencing occurred despite robust $\Delta[Ca^{2+}]_i$ at higher $[Ca^{2+}]_e$ and even if synaptic neighbors of a terminal exhibited persistent $Ca^{2+}$ entry at a lower concentration (*Figure 2A and B*), suggesting the impact of $[Ca^{2+}]_e$ on neuronal function is exerted at the single synapse level. To further evaluate whether silencing may be driven by changes in the excitability of individual branches of neurons, we expressed GCaMP6f in the cytosol to clearly label the axon and marked terminals by co-expression of mRuby-synapsin (*Figure 2—figure supplement 1A, B*). We limited our analysis to regions unambiguously representing a single branch that had at least 10 terminals to provide a robust sample. These experiments showed that individual axonal branches have both silent and responding terminals (*Figure 2—figure supplement 1C, D*), indicating that branch point failure cannot account for $[Ca^{2+}]_e$ mediated silencing.

Across a population of neurons (*Figure 3*), we found substantial heterogeneity in the proportion of silent terminals (coefficient of variation was 38%, 56%, and 57% for $[Ca^{2+}]_e$ 0.8, 1.2, and 2.0 mM, respectively) but a strikingly disproportionate increase in the mean proportion of silencing below 1.2 mM (*Figure 3C*). The change in percentage of silencing versus change in $[Ca^{2+}]_e$ was 48.9% mM$^{-1}$ transitioning from 1.2 to 0.8 mM compared to 19.0% mM$^{-1}$ from 2.0 to 1.2 mM (*Figure 3C*). The change in percentage of silencing versus change in $[Ca^{2+}]_e$ was 48.9% mM$^{-1}$ transitioning from 1.2 to 0.8 mM compared to 19.0% mM$^{-1}$ from 2.0 to 1.2 mM (*Figure 3C*). We separately tested the proportion of silencing when transitioning from 1.2 mM to 0.4 mM, observing that nearly all terminals (93%) are silent in $[Ca^{2+}]_e$ 0.4 mM (*Figure 3—figure supplement 1*). Here, the coefficient of variation, 5%, was substantially lower (32% in paired measurements in $[Ca^{2+}]_e$ 1.2 mM), demonstrating this level of $[Ca^{2+}]_e$ potently and consistently shuts down presynaptic function. The change in percentage of silencing

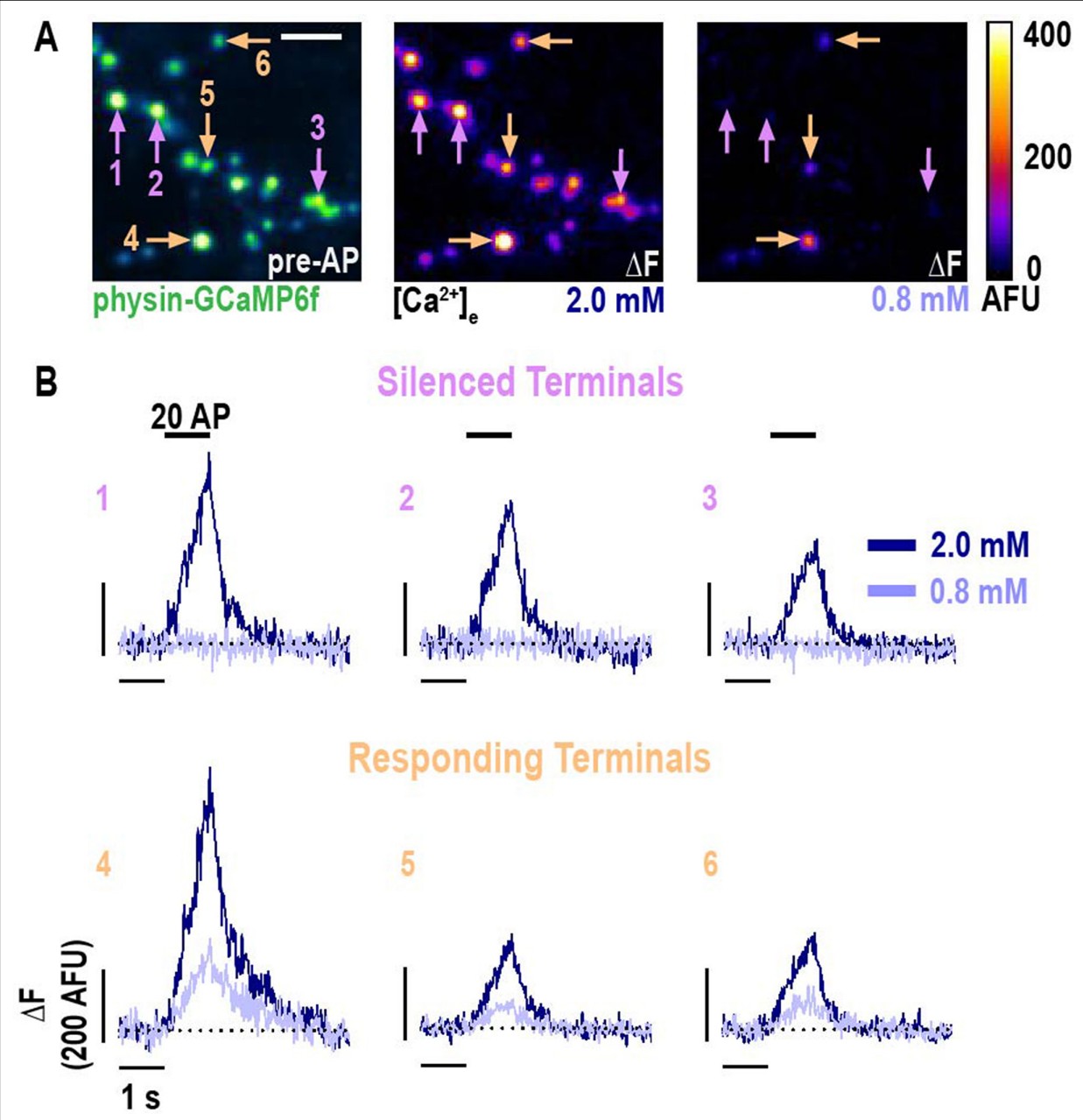

**Figure 2.** Selective nerve terminal silencing is caused by changing $[Ca^{2+}]_e$ around the physiologic set point. (**A**) Pre-stimulus average (left) and difference images (right) of AP stimulation (20 APs in 1 s) of a neuron expressing physin-GCaMP. The difference image is the mean peak (five frames) subtracted by the mean baseline (49 frames) at $[Ca^{2+}]_e$ 2.0 and 0.8 mM. For display, a subset of representative terminals was selected and a Gaussian convolution with the radius of 1 pixel was applied. (**B**) Traces of the terminals indicated in A, shown as ΔF, demonstrate selective silencing (upper row compared to lower row) of a subset of terminals. Dotted line is the pre-stimulus mean fluorescence for each terminal.

The online version of this article includes the following figure supplement(s) for figure 2:

**Figure supplement 1.** Distinctly different ΔF distributions of silent and responding nerve terminals, demonstrating $[Ca^{2+}]_e$ invariance of silent terminals.

**Figure supplement 2.** Neuronal branches contain both silent and responding nerve terminals.

versus change in $[Ca^{2+}]_e$ transition between 0.8 mM to 0.4 mM is 73.3% mM$^{-1}$, again highlighting the steep dependence of silencing on $[Ca^{2+}]_e$ below the physiologic $[Ca^{2+}]_e$. As expected, responding terminals demonstrated $\Delta[Ca^{2+}]_i$ that scaled with $[Ca^{2+}]_e$ (*Figure 3D*). Nerve terminal silencing was reversible as the proportion of silent terminals before and after switching $[Ca^{2+}]_e$ from 2.0 mM to 0.8 mM was unchanged (*Figure 3—figure supplement 2A, B*).

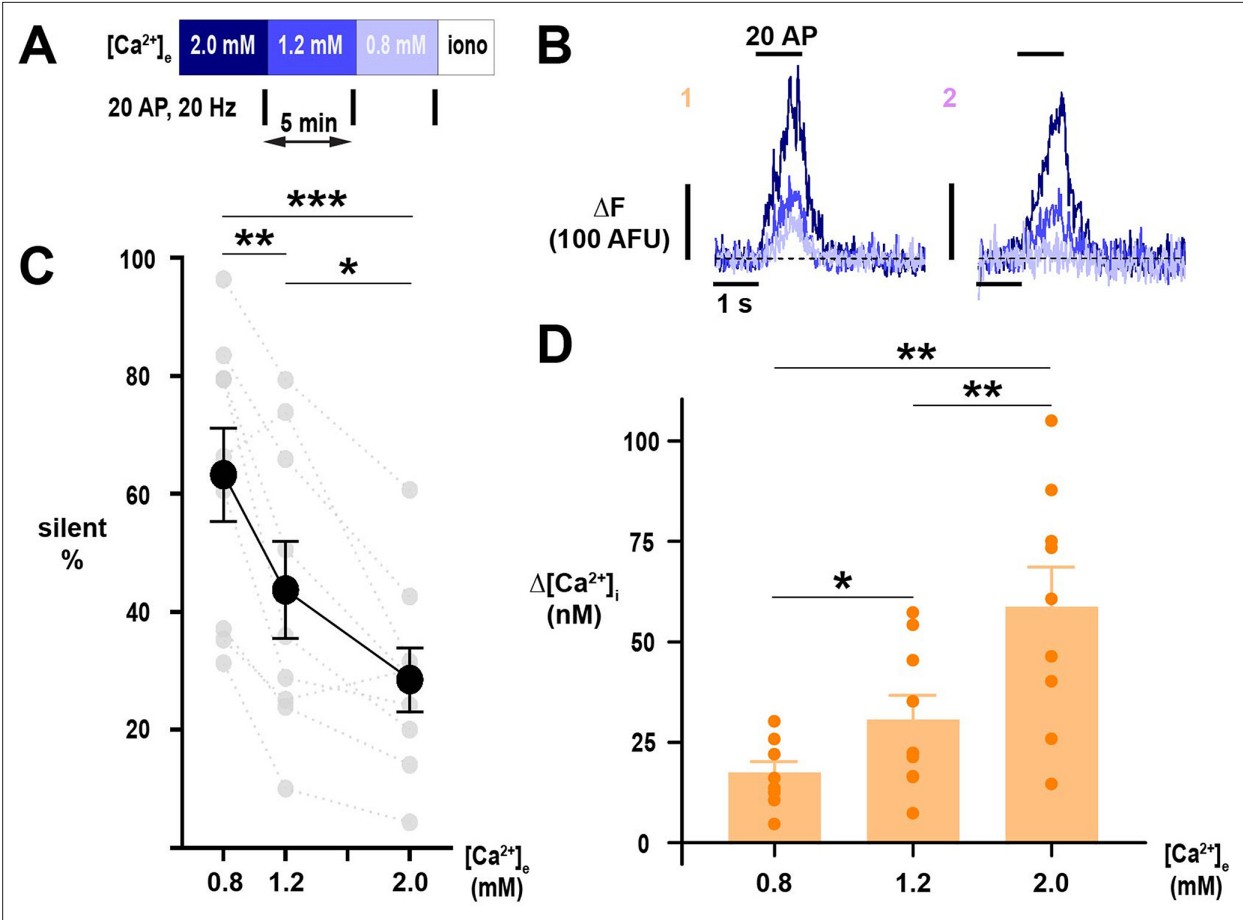

**Figure 3.** Silencing of $Ca^{2+}$ influx into nerve terminals is potently modulated by changes in $[Ca^{2+}]_e$ about the physiologic set point. (**A**) Diagram of the experimental protocol. (**B**) Example traces of single nerve terminals expressing physin-GCaMP, demonstrating persistent responsiveness (1) or silencing (2) with lowering of $Ca^{2+}$ below physiologic extracellular levels. Dotted line is pre-stimulus mean fluorescence ($\Delta F=0$). (**C**) The percentage of silent nerve terminals. Black dots are mean, error bars are SEM, and gray dots are individual cells. (**D**) Summary of $\Delta[Ca^{2+}]_i$ of responding terminals. Dots are individual cells, bars are mean, and error bars are SEM. (**C**) and (**D**) were analyzed with one-way ANOVA and Tukey's post-test for multiple comparisons, *$p<0.05$, **$p<0.01$, ***$p<0.001$, n=9.

The online version of this article includes the following figure supplement(s) for figure 3:

**Figure supplement 1.** Quantification of nerve terminal silencing in $[Ca^{2+}]_e$ 0.4 mM.

**Figure supplement 2.** Silencing of nerve terminals by lowering $[Ca^{2+}]_e$ is reversible.

## The impact of $[Ca^{2+}]_e$ on synaptic silencing is exerted through the global reduction in $Ca^{2+}$ entry irrespective of VGCC subtype

Because silencing of $Ca^{2+}$ influx occurred in a variable proportion of terminals in each neuron, we next investigated whether this selectivity is mediated by the relative expression of VGCC subtypes. $Ca^{2+}$ influx at hippocampal nerve terminals occurs predominantly via N- and P/Q-type VGCCs, with substantial heterogeneity between neurons in their proportional contribution to $Ca^{2+}$ influx (*Ariel et al., 2012*; *Ermolyuk et al., 2013*; *Hoppa et al., 2012*). Previously, cyclin-dependent kinase 5 (CDK5) was shown to silence a subpopulation of nerve terminals, an effect mediated by N-type channels only (*Kim and Ryan, 2013*). We, therefore, hypothesized that N-type channels may exhibit a greater propensity to $[Ca^{2+}]_e$-driven silencing as compared to P/Q-type channels, accounting for neuronal variability in the proportion of silent terminals. To examine whether $[Ca^{2+}]_e$-mediated silencing exhibits subtype selectivity, we utilized $\omega$-conotoxin-GVIA and $\omega$-agatoxin IVA, potent toxins that specifically inhibit N- and P/Q-type channels, respectively (*Figure 4A*). The R-type toxin, SNX-482, was employed in these experiments to exclude the minor but possibly confounding contribution of these channels to overall $Ca^{2+}$ influx (*Ermolyuk et al., 2013*). As previously demonstrated, the contribution of presynaptic VGCC

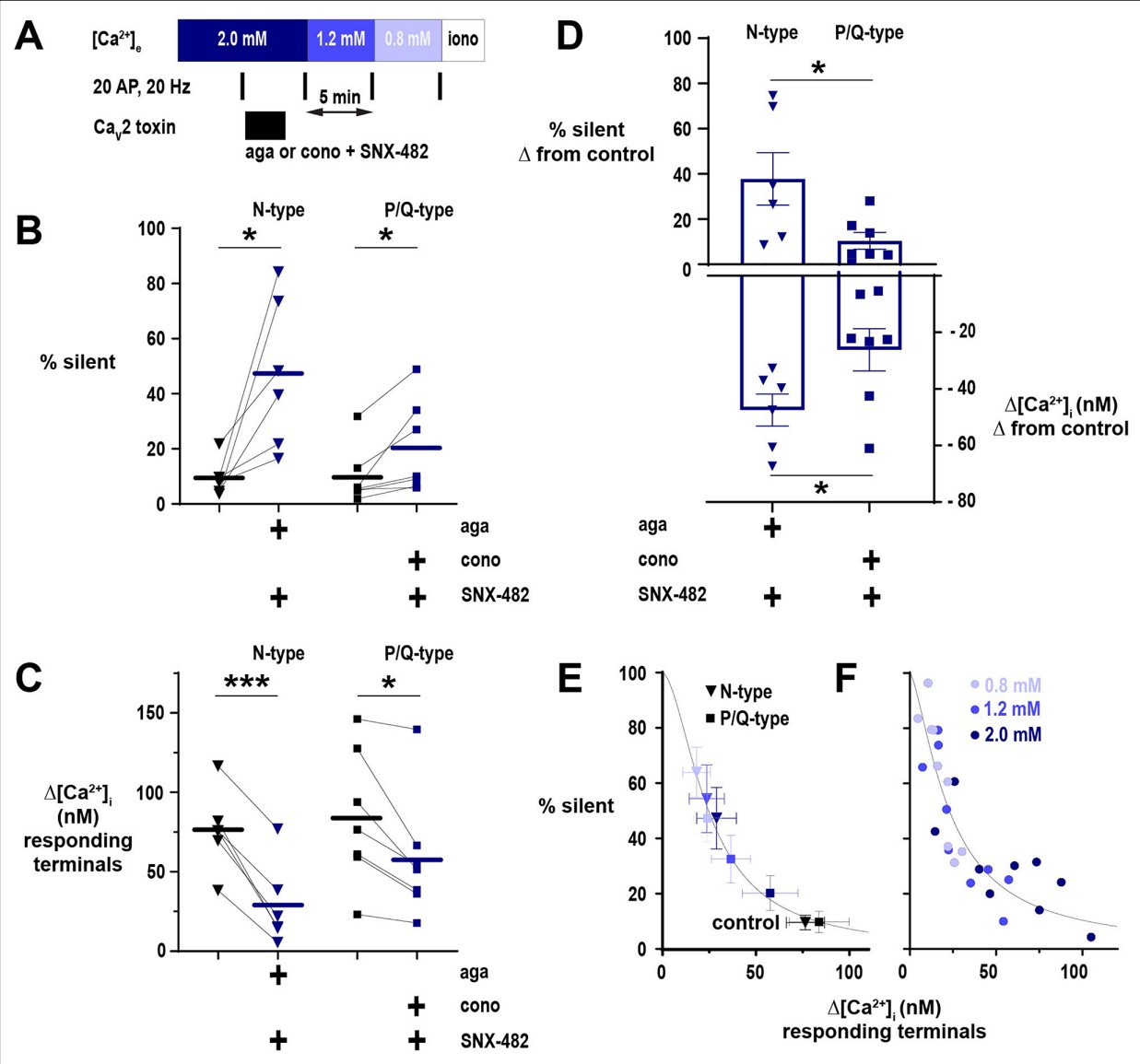

**Figure 4.** The proportion of nerve terminal silencing in neurons is related to absolute calcium (Ca²⁺) influx irrespective of Ca_V2 subtype. (**A**) Diagram of the experimental protocol applied to neurons expressing physin-GCaMP. (**B**) The percentage of silent nerve terminals before and after toxin application to isolate a Ca_V2 subtype at $[Ca^{2+}]_e$ 2 mM. (**C**) Summary of $\Delta[Ca^{2+}]_i$ in responding terminals after toxicologic isolation of N-type or P/Q-type channels. (**D**) Comparison of the effects of toxin application to isolate Ca_V2 subtypes on silencing (upper graph, shown on the left axis) and Ca²⁺ influx (lower graph, shown on the right axis). For B-D, symbols are the mean of individual neurons, lines and bars are the mean of all cells, and error bars are SEM. (**B**) and (**C**) were analyzed with a paired *t*-test, and D was analyzed with unpaired *t*-test. *p<0.05, ***p<0.001, n=6 for N-type and n=7 for P/Q-type. (**E**) The percentage of silent terminals plotted against $\Delta[Ca^{2+}]_i$ in responding terminals in control (black symbols) and following toxin application at varying $[Ca^{2+}]_e$. Symbols are mean and error bars are SEM. Data are fit with a Hill model, with the maximum at $\Delta[Ca^{2+}]_i = 0$ constrained to 100%. (**F**) Relationship of the percentage of silent terminals to $\Delta[Ca^{2+}]_i$ in responding terminals with varying $[Ca^{2+}]_e$ in neurons not treated with Ca_V2 toxins. Dots are the mean of individual cells from neurons presented in *Figure 2*. Data are fit as in E.

subtypes to $\Delta[Ca^{2+}]_i$ is variable between hippocampal neurons (*Figure 4B and C*; *Ariel et al., 2012*). A subset of terminals is silenced with acute inhibition of either subtype, but the isolation of N-type channels leads to greater proportions of silent terminals (*Figure 4B and D*). To further investigate whether a difference exists between subtypes in the likelihood of silencing, we compared Ca²⁺ entry in responding terminals to the proportion of silent terminals at each tested $[Ca^{2+}]_e$ (*Figure 4C and D*). This analysis reveals a non-linear relationship, such that lowering of absolute Ca²⁺ influx steeply increases the fraction of silent terminals. The comparison of $\Delta[Ca^{2+}]_i$ to silencing was well described with a Hill equation ($r^2 = 0.99$, $K_d = 25$ nM, $n = -1.79$) and closely matched the relationship observed

without toxin application ($r^2$ = 0.77, $K_d$ = 23 nM, n = –1.46; *Figure 4E*). This analysis indicates that the proportion of silent nerve terminals is critically related to global absolute $Ca^{2+}$ influx and that differences between N- and P/Q-type channels are driven primarily by the impact that acute inhibition has on $Ca^{2+}$ entry.

### Changing $[Ca^{2+}]_e$ near the physiologic set point potently controls the silencing of SV recycling and glutamate exocytosis at neurotransmitter release sites

Our observation of steep changes in the proportion of nerve terminals in which $Ca^{2+}$ influx was silenced by lowering $[Ca^{2+}]_e$ below the physiologic set point led us to investigate the impact of $[Ca^{2+}]_e$

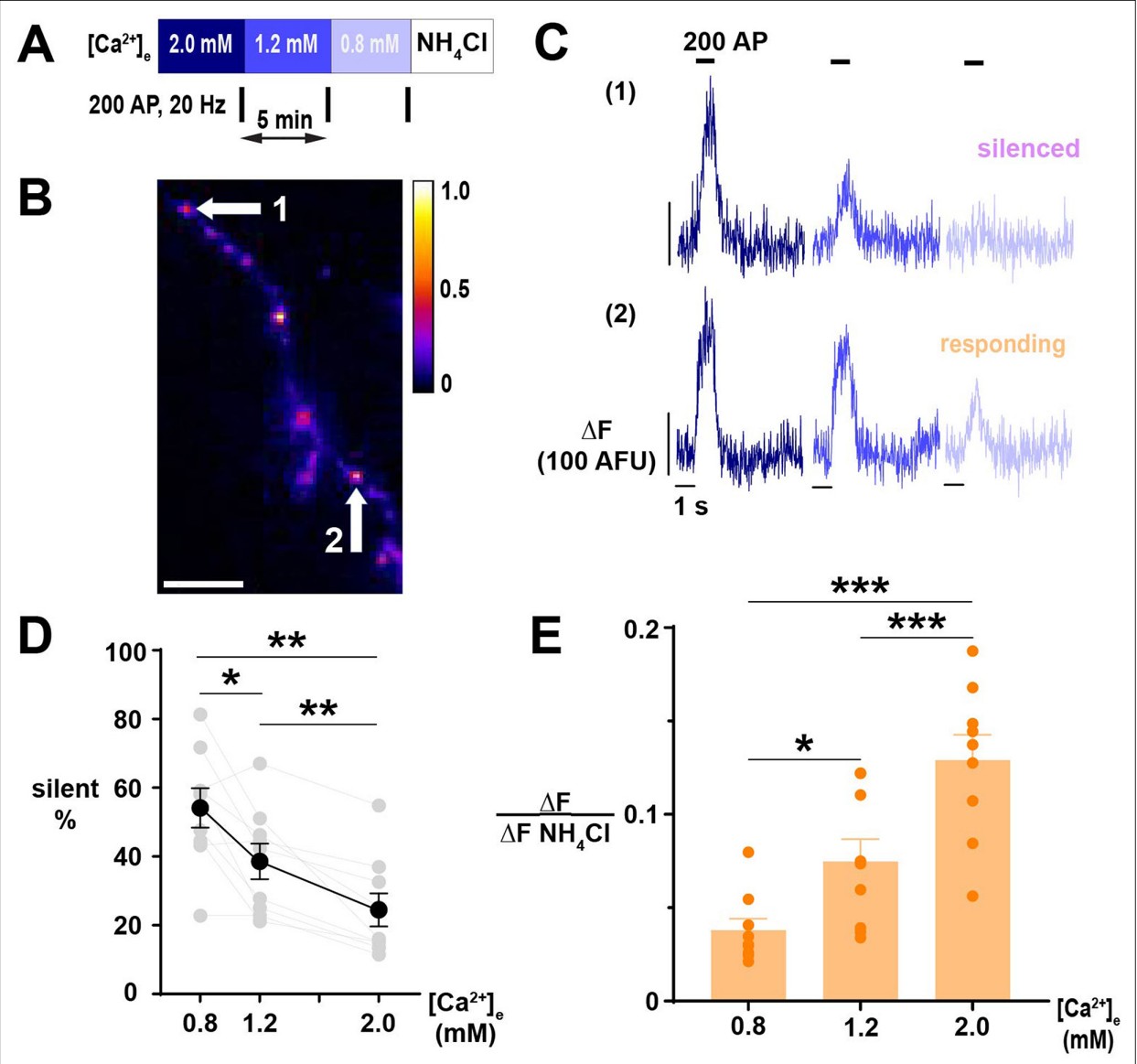

**Figure 5.** Exocytosis of SVs exhibits $[Ca^{2+}]_e$-driven silencing. (**A**) Diagram of the experimental protocol. (**B**) Segment of an axon expressing vGpH, showing a difference in fluorescence in nerve terminals revealed by the application of $NH_4Cl$ 50 mM. Scale bar is 10 μm, and the calibration is maximum-normalized fluorescence intensity. (**C**) SV recycling of individual nerve terminals indicated in B demonstrating selective silencing at $[Ca^{2+}]_e$ 0.8 mM in (1) compared to the persistent response in (2). Traces of different $[Ca^{2+}]_e$ are color-coded as in A. (**D**) Percentage of silent nerve terminals. Black dots are mean, error bars are SEM, and gray dots are individual cells. (**E**) SV exocytosis of responding terminals in different $[Ca^{2+}]_e$. Dots are individual neurons, bars are the mean, and error bars are SEM. (**D**) and (**E**) were analyzed with one-way ANOVA and Tukey's post-test for multiple comparisons, *p<0.05, **p<0.01, ***p<0.01, n=9.

on neurotransmitter release. We first addressed silencing of SV recycling with pHluorin, a pH-sensitive fluorescent protein (*Miesenböck et al., 1998*; *Sankaranarayanan et al., 2000*), tagged to vGLUT1 (vGpH), which is quenched in the acidic lumen of synaptic vesicles but fluoresces with exocytosis and exposure to the relatively alkaline extracellular environment (*Voglmaier et al., 2006*; *Figure 5A–C*). We employed a robust stimulus of 200 AP (20 Hz) to distinguish responding from silent terminals (*Kim and Ryan, 2013*; *Kim and Ryan, 2009*; *Figure 5A and C*). Similar to synaptic $Ca^{2+}$ influx, silencing of SV recycling occurred at a substantial subset of nerve terminals (~25 to 55%, *Figure 5D*). The proportion of silent terminals was heterogeneous across neurons but the mean and range at each $[Ca^{2+}]_e$ were similar to the degree of silencing observed with $\Delta[Ca^{2+}]_i$ (*Figure 3C*). As expected, the increase in SV recycling in responding terminals scaled with $[Ca^{2+}]_e$ (*Figure 5E*). Thus, as with $Ca^{2+}$ influx, SV recycling is modulated in an all-or-none manner by modest changes in $[Ca^{2+}]_e$ around the physiological set point ($[Ca^{2+}]_e$ = 1.2 mM).

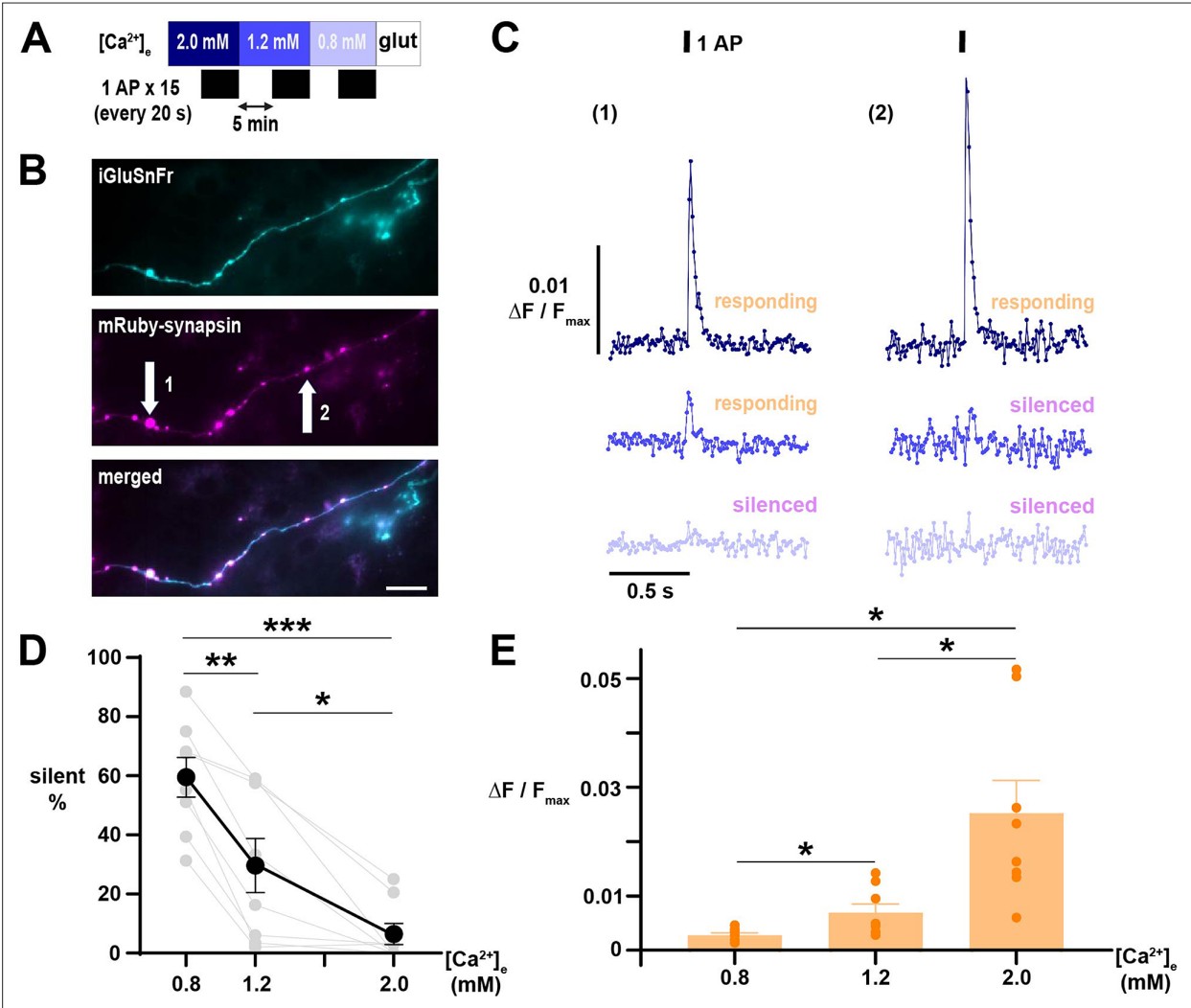

**Figure 6.** Glutamate release elicited by 1 AP demonstrates $[Ca^{2+}]_e$-driven silencing at neurotransmitter release sites. (**A**) Diagram of the experimental protocol. Responses of iGluSnFR-expressing neurons were averaged over 15 single AP trials delivered in ~20 s intervals. (**B**) Difference image (peak minus baseline fluorescence) of an axon expressing iGluSnFR (cyan, upper) stimulated with 1 AP in $[Ca^{2+}]_e$ 1.2 mM, with terminals marked by mRuby-synapsin (magenta, middle; merge lower). Scale bar 20 μm. (**B**) Traces of iGluSnFR measured at nerve terminals indicated in B, illustrating differential silencing as $[Ca^{2+}]_e$ is lowered. Traces of different $[Ca^{2+}]_e$ are color-coded as in A. (**D**) Percentage of silent nerve terminals. Black dots are mean, error bars are SEM, and gray dots are individual cells. (**E**) Glutamate release, quantified as $\Delta F/F_{max}$ in responding terminals as $[Ca^{2+}]_e$ is decreased. Dots are the mean of individual cells, bars are mean, and error bars ± SEM. (**D**) and (**E**) were analyzed with one-way ANOVA and Tukey's post-test for multiple comparisons. *p<0.05, **p<0.01, ***p<0.001, n=8.

We next sought to directly assess how $[Ca^{2+}]_e$ regulates the silencing of glutamate release in the regime of stimulation with a single AP. To do so, we utilized iGluSnFR3 v857 (iGluSnFR), a genetically-encoded, fluorescent biosensor of extracellular glutamate concentrations with an excellent signal-to-noise ratio and temporal resolution enabling quantification of glutamate release from one AP at individual nerve terminals (*Aggarwal et al., 2022*). Co-expression of mRuby-synapsin allowed measurements to be localized to neurotransmitter release sites (*Figure 6B*); however, because iGluSnFR is anchored by glycophosphatidylinositol (GPI) to the cell surface, the indicator does not necessarily discriminate between glutamate released by the transfected cell or neighboring terminals from non-transfected neurons (*Aggarwal et al., 2022*). Accordingly, while fluorescent signals at mRuby-synapsin puncta may receive contributions from additional terminals, silent terminals are definitively assigned as inactive. With this approach, we found glutamate exocytosis evoked by a single AP exhibited higher proportions of silencing at neurotransmitter release sites as $[Ca^{2+}]_e$ was lowered, again demonstrating a marked increase in silencing as the concentration decreased below the physiologic set point (*Figure 6C and D*). Thus, using these two fluorescent biosensors quantifying neurotransmitter handling under different stimulation regimes demonstrates that $[Ca^{2+}]_e$ is an important regulator of presynaptic function, differentially dictating across synapses and neurons whether neurotransmitter will be released.

## GPCRs that lower synaptic $Ca^{2+}$ currents similarly modulate the proportion of silent nerve terminals

The selective silencing of $Ca^{2+}$ entry and neurotransmitter release in subpopulations of nerve terminals by globally reducing $Ca^{2+}$ influx may be an important mechanism determining the behavior of synapses to biological processes affecting $Ca^{2+}$ currents ($I_{Ca}$). For instance, a subset of important presynaptic GPCRs affects neurotransmission by lowering synaptic $Ca^{2+}$ influx (*Dolphin, 2003*). One such canonical mechanism is the agonism of the $GABA_B$ receptor ($GABA_BR$), which lowers $I_{Ca}$ by approximately 50% via interactions of the $G\beta\gamma$ subunit with presynaptic VGCCs (*Mintz and Bean, 1993*; *Herlitze et al., 1996*; *Isaacson, 1998*; *Laviv et al., 2011*). Thus, we hypothesized that $GABA_BR$-mediated

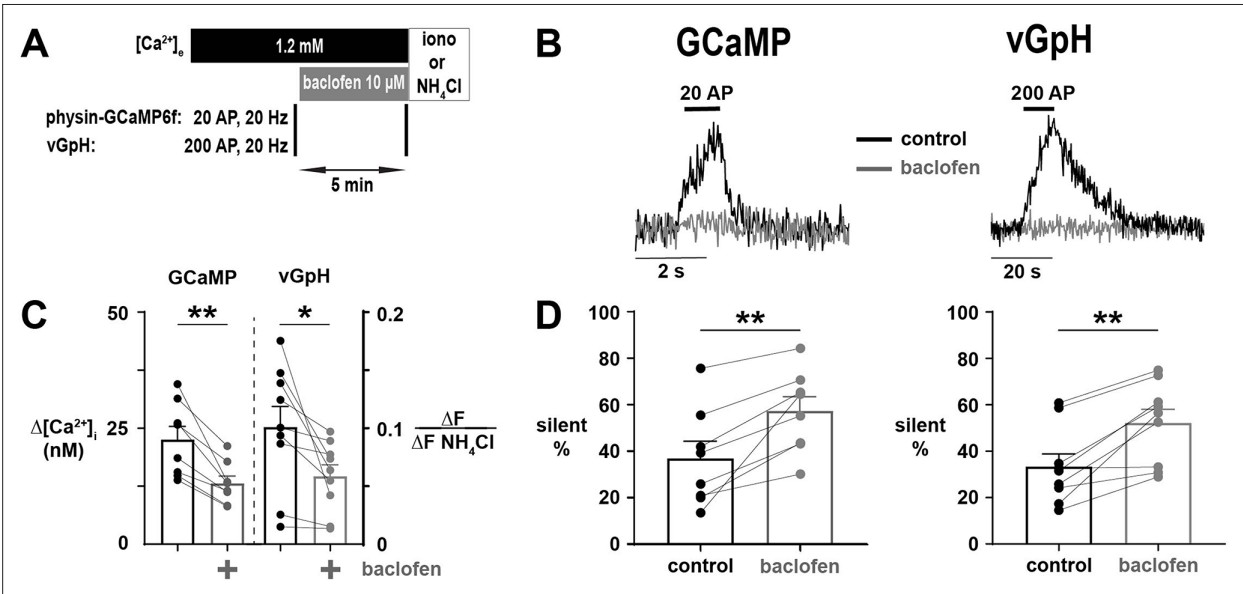

**Figure 7.** Agonism of $GABA_BR$ with baclofen leads to silencing of calcium ($Ca^{2+}$) influx and SV exocytosis. (**A**) Diagram of the experimental protocol. (**B**) Representative traces of silencing of $Ca^{2+}$ influx and SV exocytosis resulting from agonism of $GABA_BR$ with baclofen in single nerve terminals expressing physin-GCaMP or vGpH, respectively. (**C**) $\Delta[Ca^{2+}]_i$ and $\Delta F/\Delta F_{NH4Cl}$ before and following the application of baclofen in responding terminals. (**D**) The percentage of silent terminals before and following treatment with baclofen was measured in physin-GCaMP and vGpH-expressing neurons (see labels above). For (**C**) and (**D**), dots are individual neurons, bars are mean of all neurons, and error bars are SEM. (**C**) and (**D**) were analyzed with a paired t-test, *p<0.05, **p<0.01, n=8 for physin-GCaMP, and n=9 for vGpH.

The online version of this article includes the following figure supplement(s) for figure 7:

**Figure supplement 1.** Agonism of $GABA_BR$ with baclofen causes fewer terminals to become silent in $[Ca^{2+}]_e$ 2.0 mM compared to 1.2 mM.

decreases in $I_{Ca}$ will significantly increase the degree of nerve terminal silencing at the physiologic set point of $[Ca^{2+}]_e$. We compared the effect of baclofen, a $GABA_BR$ agonist, on the silencing of $Ca^{2+}$ influx and SV recycling using physin-GCaMP and vGpH, respectively (*Figure 7A–C*). Consistent with our proposed mechanism, agonizing $GABA_BR$ at $[Ca^{2+}]_e$ 1.2 mM caused an additional ~20% of terminals to become silent in both $Ca^{2+}$ influx and SV recycling (*Figure 7D*), similar to the proportional increase we observed decreasing $[Ca^{2+}]_e$ to 0.8 mM. However, agonism with baclofen in $[Ca^{2+}]_e$ 2.0 mM led to a substantially lower increase in the proportion of silencing (7.6 ± 3.4% vs 20.5 ± 4.8%, p=0.017; *Figure 7—figure supplement 1A, B*). Thus, GPCR-mediated silencing was dependent on $[Ca^{2+}]_e$ such that the impact was far greater operating at the physiologic $[Ca^{2+}]_e$. These results confirm that GPCR-mediated modulation acting presynaptically to decrease $I_{Ca}$ can silence a substantial proportion of nerve terminals.

## Discussion

The flux of $Ca^{2+}$ in neurons plays a central role in neurotransmission by triggering the exocytosis of SVs, thereby releasing neurotransmitters into the synaptic cleft (*Dittman and Ryan, 2019*). Uncovering the mechanisms regulating $Ca^{2+}$ entry into nerve terminals is therefore critical to understanding the molecular underpinnings of CNS function. The relationship between $Ca^{2+}$ influx and the exocytotic rate is highly cooperative (*Heidelberger et al., 1994*; *Schneggenburger and Neher, 2000*; *Bollmann et al., 2000*; *Ariel and Ryan, 2010*), highlighting the amplification of neurotransmitter release caused by changes in $\Delta[Ca^{2+}]_i$. Fewer studies have sought to determine the effect of $[Ca^{2+}]_e$ on $\Delta[Ca^{2+}]_i$ and, to our knowledge, none have examined this in the range encompassing the physiological set point for $[Ca^{2+}]_e$ at physiological temperature. In general, the narrowing of the AP at warmer temperatures (*Hodgkin and Katz, 1949*; *Yu et al., 2012*) results in lowering of $Ca^{2+}$ influx during AP firing, so additionally studying lower $[Ca^{2+}]_e$ results in a much lower regime of $\Delta[Ca^{2+}]_i$. Previous attempts (carried out at sub-physiological temperatures) modeled this relationship with Michaelis-Menten kinetics, where $\Delta[Ca^{2+}]_i$ obeyed a simple saturation of VGCCs, but predicted a simple 1:1 relationship extrapolating to lower $[Ca^{2+}]_e$ (*Ariel and Ryan, 2010*; *Schneggenburger et al., 1999*).

Here, we show an unexpected and impactful behavior of $Ca^{2+}$ influx in the lower $\Delta[Ca^{2+}]_i$ regime, which is that small modulation of $Ca^{2+}$ influx leads to dramatic silencing of individual nerve terminals. We discovered that a substantial subset of terminals (~40%) exhibited silencing of $Ca^{2+}$ influx and SV exocytosis at physiologic $[Ca^{2+}]_e$, and this proportion increased steeply as $[Ca^{2+}]_e$ was lowered to 0.8 mM and 0.4 mM (to ~60% and >90%, respectively). In comparison, the change in the proportion of silent terminals in the transition from $[Ca^{2+}]_e$ 2.0 mM to 1.2 mM, a difference of ~15%, was smaller despite a larger step in $[Ca^{2+}]_e$. Previous estimates of synaptic silencing were drawn from experiments conducted at $[Ca^{2+}]_e$ 2.0 mM and agree with our findings at this concentration (*Altrock et al., 2003*; *Kim and Ryan, 2013*; *Moulder et al., 2004*; *Moulder et al., 2006*). However, our results highlight that the proportion of silent terminals in cultured neurons is substantially higher under conditions of physiologic $[Ca^{2+}]_e$. Thus, $[Ca^{2+}]_e$ is a crucial variable to consider in future studies examining synaptic silencing. The agreement of our findings in the silencing of both $\Delta[Ca^{2+}]_i$ and SV exocytosis indicate that silencing of $\Delta[Ca^{2+}]_i$ is driving the effect as opposed to an independent mechanism operating on SV exocytosis alone.

The proportion of silencing across different neurons was steeply related to absolute $\Delta[Ca^{2+}]_i$, such that lowering of $\Delta[Ca^{2+}]_i$ was associated with increased proportions of silent terminals. Interestingly, this relationship persisted despite the selective blockade of VGCC subtypes by potent toxins, demonstrating the propensity for silencing is not related to their relative distribution at nerve terminals. Due to this effect, our analysis suggests that the minimal $[Ca^{2+}]_e$ needed to sustain any neurotransmission is ~0.4–0.5 mM. Moreover, the degree of silencing caused by acutely blocking VGCC subtypes followed this relationship. Taken together, our results suggest the operation of a feedback mechanism causing the selective shutdown of terminals not meeting a threshold for $Ca^{2+}$ entry. This mechanism has important implications for the impact of neuromodulators that regulate $I_{Ca}$ because relatively small changes in $\Delta[Ca^{2+}]_i$ could cause a substantial proportion of terminals to become silent (*Dolphin, 2003*). As an example, we investigated the impact on synaptic function resulting from the inhibition of presynaptic VGCCs by $G\beta\gamma$ subunits using agonism of $GABA_BR$ by baclofen. Previous studies have demonstrated that the $G\beta\gamma$ interacts with presynaptic VGCCs to lower $I_{Ca}$ (*Isaacson, 1998*; *Laviv et al., 2011*; *Kajikawa et al., 2001*). We observed that, in responding terminals, baclofen

was associated with ~40% decrease in $\Delta[Ca^{2+}]_i$. This decrement in $Ca^{2+}$ influx led to an increase in the proportion of silent terminals from ~40 to 60%. Notably, this difference in the degree of silencing and $\Delta[Ca^{2+}]_i$ agreed excellently with results from decreasing $[Ca^{2+}]_e$ 1.2–0.8 mM, supporting the conclusion that the effect of baclofen is mediated by decreasing presynaptic $Ca^{2+}$ influx. In addition, baclofen caused substantially fewer terminals to become silent in $[Ca^{2+}]_e$ 2.0 mM, highlighting the importance of physiologic $[Ca^{2+}]_e$ in dictating synaptic function.

While our findings have shown that $[Ca^{2+}]_e$ contributes importantly to presynaptic neuromodulation, we have not identified the precise molecular mechanism. Previous investigations indicate that silencing is a drastic form of synaptic homeostasis, as chronic suppression of neuronal activity with tetrodotoxin (TTX) decreases the proportion of silent terminals (*Moulder et al., 2006*) and increasing activity with chronic depolarization increases this proportion (*Moulder et al., 2004*; *Moulder et al., 2006*). Several intracellular signaling pathways have been identified to set the fraction of silent terminals, including those involving adenylyl cyclase (*Moulder et al., 2008*), calcium/calmodulin-dependent protein kinase II (*Ninan and Arancio, 2004*), and calcineurin/cyclin-dependent kinase 5 (*Kim and Ryan, 2010*; *Kim and Ryan, 2013*). The disparate molecular pathways that converge on the regulation of synaptic silencing suggest that it is an important mode of plasticity for neurons in the CNS. Notably, in experiments investigating the impact of GPCR modulation, only a subset of GPCRs tested was able to alter silencing (*Crawford et al., 2011*). The silencing caused by GABA$_B$R agonism was dependent on proteasome function, and the authors concluded that protein degradation was necessary (*Crawford et al., 2011*). However, these experiments utilized 4 hr incubation with baclofen, as opposed to 5 min here, so an acute effect of GABA$_B$R agonism independent of proteasomal activity may have been missed. Indeed, silencing may have both proteasome-dependent and independent pathways, operating on slow (hours) or acute (<1 hr) time scales, respectively (*Crawford et al., 2012*). Moreover, pharmacologic inhibition of proteasome function prevents depolarization-induced silencing, so the effect of proteasome inhibition may not be specific to GABA$_B$R agonism (*Jiang et al., 2010*). Thus, our results suggest a separate, rapid mechanism by which nerve terminals can be silenced, adding to the diversity of pathways by which this form of plasticity is achieved.

We classified terminals for which the $\Delta F$ of $Ca^{2+}$ entry falls below one standard deviation of pre-stimulus fluorescence as silent. This previously used definition agrees well with the other approaches used (*Moulder et al., 2008*; *Moulder et al., 2004*; *Moulder et al., 2006*; *Crawford et al., 2011*; *Crawford et al., 2012*; *Hogins et al., 2011*). Given that $Ca^{2+}$ influx is itself non-linearly related to SV exocytosis, even if $Ca^{2+}$ entry has only been inhibited to undetectable levels, this level would lead to even rarer neurotransmitter release. The sub-proportional relationship of $\Delta[Ca^{2+}]_i$ to $[Ca^{2+}]_e$ and the heterogeneous changes of individual nerve terminals to changes in $[Ca^{2+}]_e$ refutes the simple biophysical model of the relationship of $Ca^{2+}$ influx to $[Ca^{2+}]_e$. Rather, our results support nerve terminal specificity in the propensity to have presynaptic function diminished by perturbations that lower the driving force for $Ca^{2+}$ entry.

Our experiments have also not assessed whether silencing occurs independently at individual release sites. The sensor iGluSnFr3 can resolve spontaneous glutamate exocytosis that represents a single vesicle and, therefore, the output of a single release site (*Aggarwal et al., 2022*). However, quantifying the proportion of release sites that are silenced is limited by the variable number of active zones per nerve terminal in a population of CA1-CA3 hippocampal neurons (*Schikorski and Stevens, 1997*; *Rigby et al., 2022*). Moreover, iGluSnFr can detect glutamate released from neighboring but non-transfected neurons (*Aggarwal et al., 2022*). Thus, it would be difficult with our approach to unambiguously identify a release site and, therefore, to quantify the proportion of sites silenced. We speculate that because $Ca^{2+}$ influx at the terminal is prevented, all active zones within a terminal are equally impacted by $[Ca^{2+}]_e$ mediated silencing.

Our findings highlight that the physiologic set point of $[Ca^{2+}]_e$ in the CNS functions as an important lever in the modulation of synaptic function because further lowering of $\Delta[Ca^{2+}]_i$ substantially increases the proportion of silent terminals. Although $[Ca^{2+}]$ of the CSF is tightly regulated globally (*Forsberg et al., 2019*; *Jones and Keep, 1988*; *Barkai and Meltzer, 1982*), models of the synapse that account for its limited volume and restriction of diffusion have predicted that drastic reductions of local $[Ca^{2+}]_e$ may occur during repetitive presynaptic firing (*Egelman and Montague, 1999*; *King et al., 2001*). Thus, the phenomenon we observed may be important in vivo as a homeostatic mechanism to prevent

excessive or runaway synaptic activity (*Stringer et al., 2007*). Our results suggest that the magnitude of the decrease in $[Ca^{2+}]_e$ does not need to be large to shut down a sizable proportion of terminals.

In summary, we have utilized highly sensitive, genetically-encoded fluorescent biosensors to dissect the effects of $[Ca^{2+}]_e$ near the physiologic set point on presynaptic function. We discovered that $[Ca^{2+}]_e$ has a potent effect in setting the proportion of silent terminals which is driven by $\Delta[Ca^{2+}]_i$. These findings provide evidence that $[Ca^{2+}]_e$ is an important lever contributing to neuromodulation. Future studies will address the intracellular molecular targets responsible for this acute mechanism of synaptic silencing.

# Materials and methods

## Animals

All experiments involving animals were performed in accordance with protocols approved by the Weill Cornell Medicine Institutional Animal Care and Use Committee. Neurons were derived from Sprague-Dawley rats (Charles River Laboratories strain code: 001, RRID: RGD_734476) of either sex on postnatal days 0–2.

## Neuronal culture

Primary neuronal cultures were prepared as previously described (*Ryan, 1999*). Hippocampal CA1 to CA3 regions were dissected, dissociated, and plated onto poly-L-ornithine-coated coverslips. Plating media consisted of the minimal essential medium, 0.5% glucose, insulin (0.024 g/l), transferrin (0.1 µg/l), GlutaMAX 1%, N-21 (2%), and fetal bovine serum (10%). After 1–3 days in vitro (DIV), cells were fed and maintained in media with the following modifications: cytosine β-D-arabinofuranoside (4 µM) and FBS 5%. Cultures were incubated at 37 °C in a 95% air/5% $CO_2$ incubator. Calcium phosphate-mediated gene transfer was performed on DIV 6–8, and neurons were used for experiments on DIV 14–21.

## Plasmid constructs

The following published DNA constructs were used: VGLUT1-pHluorin (vGpH) (*Voglmaier et al., 2006*), synaptophysin-GCaMP6f (physin-GCaMP) (*de Juan-Sanz et al., 2017*), cytosolic GCaMP6f (*Chen et al., 2013*), and GPI iGluSnFR3 v857(iGluSnFR) (*Aggarwal et al., 2022*), which was a gift from Kasper Podgorski. mRuby3-synapsin1a (Addgene plasmid #187896) was generated by removing GFP from GFP-synapsin (*Chi et al., 2001*) using restriction sites AgeI and BGIII, and substituting it in frame with mRuby obtained from pKanCMV-mRuby3-18aa-actin, which was a gift from Michael Lin (Addgene plasmid #74255). Cytosolic GCaMP6f under the CaMKII promoter was generated by cloning GCaMP6f (*Chen et al., 2013*) (Addgene plasmid #40755) into a CaMKII promoter vector (*Chow et al., 2010*) (Addgene plasmid #22217) as previously described (*de Juan-Sanz et al., 2017*).

## Live-cell imaging

Coverslips were loaded onto a custom chamber and perfused at 100 µl min$^{-1}$ via a syringe pump (Fusion 4000, Chemyx) with Tyrode's solution containing (in mM, except if noted otherwise): NaCl 119, KCl 2.5, glucose 30, HEPES 25, $CaCl_2$ 0.8–2.0 with $MgCl_2$ adjusted to maintain divalence of 4, D, L-2-amino-5-phosphonovaleric acid (APV) 50 µM, 6-cyano-7-nitroquinoxaline-2,3-dione (CNQX) 10 µM, adjusted to pH 7.4. Temperature was maintained at 37 °C with a custom-built objective heater under feedback control (Minco). Fluorescence was stimulated with OBIS 488 nm LX or OBIS 561 nm LS lasers (Coherent) passing through a laser speckle reducer (LSR 3005 at 12° diffusion angle, Optotune). Live-cell imaging was performed with a custom-built, epifluorescence microscope. Emission was acquired with a 40 x, 1.3 numerical aperture objective (Fluar, Zeiss) and Andor iXon + Ultra 897 electron-multiplying charge-coupled device camera. APs were evoked with platinum-iridium electrodes generating 1 ms pulses of 10 V cm$^{-1}$ field potentials via a current stimulus isolator (A385, World Precision Instruments). For physin-GCaMP and vGpH, neurons were stimulated with 20 and 200 AP, respectively, delivered at 20 Hz. For iGluSnFR, neurons were stimulated with a single AP, with averaging performed over 15 trials delivered every 20 s. A custom-designed Arduino board coordinated AP and laser stimulation with frame acquisition. Frame rates for AP recordings with vGpH, GCaMP6f, and iGluSnFR were 5, 50, and 100 Hz, respectively. To achieve frame rates of 50 and 100 Hz, a subregion of the EMCCD

chip was used (347 pixels and 169 pixels, respectively, compared to 512 pixels). In experiments with neurons expressing vGpH, Tyrode's solution with NH$_4$Cl 50 mM replacing an equimolar concentration of NaCl was perfused at the end of the experiment to alkalinize intra-vesicular pHluorin molecules (*Sankaranarayanan et al., 2000*) and enable normalization of fluorescence to the total pool of internal vesicles (*Kim and Ryan, 2010*). Because vGpH has low surface accumulation and fluorescence is therefore largely quenched before AP stimulation (*Balaji and Ryan, 2007*), a brief (<1 min) exposure to NH$_4$Cl was performed before experiments to identify transfected neurons and then washed out for at least 5 min. In experiments with physin-GCaMP, the chamber perfusate was exchanged for Tyrode's solution with Ca$^{2+}$ 4 mM, pH 6.9, and ionomycin 500 μM to saturate the fluorophore and allow conversion of fluorescence to absolute [Ca$^{2+}$]i (*de Juan-Sanz et al., 2017*). In experiments with iGluSnFR, glutamate 100 mM was applied to saturate the sensor and normalize to maximum fluorescence (*Aggarwal et al., 2022*). Recordings of physin-GCaMP6f and vGpH included ~200 nerve terminals on average while recordings of iGluSnFR included ~50 nerve terminals due to the limited EMCCD dimensions at this frame rate.

## Key pharmacological reagents

Ionomycin, ω-conotoxin GVIA, ω-agatoxin IVA, and SNX-482 were purchased from Alomone labs (catalog numbers I-700, C-300, STA-500, and RTS-500 respectively). The toxins were applied for 3 min in Tyrode's solution at concentrations of 1 μM, 400 nM, and 500 nM for ω-conotoxin GVIA, ω-agatoxin IVA, and SNX-482, respectively (*Ariel et al., 2012*; *Ermolyuk et al., 2013*; *Hoppa et al., 2012*). Baclofen was purchased from Sigma-Aldrich (catalog number B 5399) and continuously perfused at 10 μM beginning 5 min before AP stimulation (*Laviv et al., 2011*).

## Image analysis

Images were analyzed with ImageJ (*Rasband, 1997*). Circular regions of interest (ROIs) were semi-automatically placed around puncta corresponding to nerve terminals using a custom-written macro. Fluorescence within ROIs was background corrected by subtraction of signal from adjacent, non-synaptic ROIs using a custom-written macro. Images of individual 1 AP trials of iGluSnFr were averaged using a custom-written macro.

## Experimental design and statistical analyses

Quantification of peak fluorescence responses and the percentage of silent terminals was performed with Excel. Terminals were classified as silent if the peak (ΔF) above the baseline (mean of 49 frames) was less than the standard deviation of the baseline (σ$_{baseline}$, 49 frames) (*Kim and Ryan, 2010*). For physin-GCaMP and vGpH, the peak was calculated as the mean of five frames at the end of the AP train. For iGluSnFR, the peak was calculated as the mean of 3 frames, with the first frame in the range coinciding with the AP. For vGpH, fluorescence was normalized to the total internal pool of vesicles following perfusion of NH$_4$Cl (*Kim and Ryan, 2010*). For iGluSnFR, fluorescence was normalized to the maximum following perfusion of a saturating concentration of glutamate (*Aggarwal et al., 2022*). For physin-GCaMP, fluorescence was converted to absolute [Ca$^{2+}$]i using (*de Juan-Sanz et al., 2017*):

$$\left[Ca^{2+}\right]_e = K_d \left( \frac{\frac{F}{F_{max}} - \frac{1}{R_f}}{1 - \frac{F}{F_{max}}} \right)^{\frac{1}{n}}$$

where F$_{max}$ is the peak following saturation of GCaMP6f with ionomycin and K$_d$ (0.38 μM), R$_f$ (51.8), and n (2.3) are the in vitro dissociation constant, dynamic range, and Hill coefficient, respectively (*Chen et al., 2013*). Statistical analysis was performed with Prism version 9. Comparisons were made with the *t*-test except for *Figure 7—figure supplement 1B* which was analyzed with the Mann-Whitney U test. For more than two groups, ANOVA with Tukey's post-test for multiple comparisons was used. Statistical significance was defined as $p < 0.05$. The relationship of Δ[Ca$^{2+}$]i in responding terminals to the proportion of silent terminals was fit with a Hill equation using least squares regression and constraining the maximum silencing at Δ[Ca$^{2+}$]i = 0 to 100% (*Figure 4E and F*). Data are presented as mean ± SEM except in *Figure 1D* and *Figure 1—figure supplement 1B*, which demonstrates a mean ± 95% confidence interval to illustrate the experimental difference from the theoretical value.

## Materials availability

Fluorescent biosensors are available through Addgene (iGluSnFR plasmid #187896, mRuby3-synapsin1a plasmid #74255) or from the corresponding author by request (physin-GCaMP, CKII cytosolic GCaMP6f, vGpH). Custom-written macros for ImageJ analysis are available through GitHub (*Cook and Ryan, 2022*). Supporting data is available from Dryad (doi:10.5061/dryad.1zcrjdfw0).

## Acknowledgements

We thank members of the Ryan lab for their valuable discussion of the manuscript. We give special thanks to Andrew Nelson for developing the custom Arduino board and its graphical user interface. We thank Kasper Podgorski for kindly providing iGluSnFR3 v857. This work was funded by NIH grants to TAR (NS036942 and NS117139), NIH grant to DCC (K08GM148935), a Mentored Research Training Grant (MRTG) from the Foundation for Anesthesia Education and Research (FAER) to DCC (MRTG-02-15-2020-Cook), and a grant from the Burroughs Wellcome Weill Cornell Physician-Scientist Academy to DCC.

## Additional information

### Funding

| Funder | Grant reference number | Author |
|---|---|---|
| NINDS | NS036942 | Timothy A Ryan |
| National Institute of General Medical Sciences | GM148935 | Daniel C Cook |
| National Institute of Neurological Disorders and Stroke | NS117139 | Timothy A Ryan |
| Foundation for Anesthesia Education and Research | MRTG-02-15-2020-Cook | Daniel C Cook |
| Burroughs Wellcome Weill Cornell Physician-Scientist Academy | | Daniel C Cook |

The funders had no role in study design, data collection and interpretation, or the decision to submit the work for publication.

### Author contributions

Daniel C Cook, Conceptualization, Data curation, Software, Formal analysis, Investigation, Visualization, Writing – original draft, Writing – review and editing; Timothy A Ryan, Conceptualization, Resources, Supervision, Funding acquisition, Investigation, Visualization, Methodology, Writing – original draft, Project administration, Writing – review and editing

### Author ORCIDs

Daniel C Cook http://orcid.org/0000-0003-0714-4050
Timothy A Ryan http://orcid.org/0000-0003-2533-9548

### Ethics

All experiments involving animals were performed in accordance with protocols approved by the Weill Cornell Medicine Institutional Animal Care and Use Committee (IACUC protocol 0601-450A).

### Decision letter and Author response

Decision letter https://doi.org/10.7554/eLife.83530.sa1
Author response https://doi.org/10.7554/eLife.83530.sa2

## Additional files

**Supplementary files**
• MDAR checklist

**Data availability**
All data generated or analyzed during this study is included in the manuscript and supporting file; Source Data has been uploaded onto Dryad (doi:10.5061/dryad.1zcrjdfw0) and customized code has been uploaded to Github (https://github.com/taryan2020/ImageJ, (*Cook and Ryan, 2022* copy archived at swh:1:rev:a33d53ca21ac4bd2fbe1afc52eb9162b683ce0a0)).

The following dataset was generated:

| Author(s) | Year | Dataset title | Dataset URL | Database and Identifier |
|---|---|---|---|---|
| Ryan TA, Cook D | 2022 | Supporting Data | https://dx.doi.org/10.5061/dryad.1zcrjdfw0 | Dryad Digital Repository, 10.5061/dryad.1zcrjdfw0 |

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
