## [Editor Report]

In this important study Cook and Ryan find that at physiological temperatures, small changes in extracellular calcium concentrations shift individual presynaptic terminals between active and inactive states. This discovery is extended to the action of inhibitory synaptic modulation via GABA, emphasizing potentially broad relevance for our understanding of how synaptic transmission is stabilized and modulated throughout the brain. The evidence for the observed phenomenon is compelling, given the use of multiple functional reporters and rigorous analyses.

---

## [Decision Letter]

**Decision letter after peer review:**

Thank you for submitting your article "Neuromodulatory silencing of nerve terminals" for consideration by *eLife*. Your article has been reviewed by 3 peer reviewers, including Graeme W Davis as the Reviewing Editor and Reviewer #1, and the evaluation has been overseen by John Huguenard as the Senior Editor.

The essential revisions are representative of conversation among reviewers, attempting to synthesize the individual reviews.

1. Two reviewers focused on a specific issue: The obvious challenge is distinguishing between a very small response and a non-responder. The data, throughout the paper, are categorized (yes/no) and presented as a percentage. Thus, the crux of the manuscript relies upon thoroughly demonstrating that the discrimination between a responder and non-responder is not confounded. Suggestions for how this might be achieved are presented, but these are not intended to be the only solution to addressing this issue – it is entirely up to the authors.

2. The data set could be broadly strengthened by including data from individual release sites, rather than simply presenting averages. This is particularly the case with the GluSnFr data where the probabilistic release of single vesicles could be resolved, inclusive of amplitudes associated with spontaneous release events (when possible).

3. Two reviewers highlight a circular argument regarding the classification of silent sites, that are then shown to be silent. This should be addressed either through changes to the figures or the text. Reviewer 3 (comment 3). Reviewer 2 (final comment).

4. It would be important, in at least one experimental paradigm, to see that release returns to baseline following restoration of external calcium.

5. The issue of neuronal excitation can and should be evaluated. It is understood that these are sparse cultures. None-the-less, it remains possible that fields of view contain axons originating in more than one neuron, or in the case that they reside on different branches of the same axon, that branch point failures could 'cause' boutons to appear to function differently. A solution would be to systematically track whether sets of ROI unambiguously residing on single branches still function differently with respect to release (some being silenced and others not a lower calcium).'

6. A core feature of the manuscript is the quantification of calcium entry via the genetically encoded calcium sensor. There was a healthy debate among reviewers regarding the necessity of utilizing synthetic dyes. On the one hand, the superior properties of the dyes allow for highly quantitative validation of existing transgenic reporters and central conclusions. On the other hand, this requires loading (likely through patch pipettes). Addressing the robustness of the existing measures, either through dyes or validation with additional variants of transgenic reporters would strengthen the work. Again, however, the precise nature of the experiment is entirely up to the authors.

7. Two reviewers, while agreeing that the GABA data are interesting, commented that the data are somewhat under-developed. On the one hand, the title emphasizes "neuromodulation" and this could be modified to better reflect the single experiment shown regarding GABA. At another level, reviewer #3 argues that "The interesting issue would be to reveal how the effect of GABAB activation depends on the [Ca^2+^]e. This information is essential to see whether there is indeed a shoulder in its effectiveness curve." The title likely should be changed, at a minimum.

8. There was discussion about the main conclusion of the manuscript. 'Thus, as with Ca^2+^ influx, SV recycling is modulated in an all-or-none manner by modest changes in [Ca^2+^]e around the physiological set point.' It seems equally likely that authors have identified a steeper continuum rather than 'all or none' control. As this is a central tenant of the manuscript, it could perhaps be addressed through changes in the text to shore up the argument, differently emphasized through data presentation or through the addition of new data.

*Reviewer #1 (Recommendations for the authors):*

While fascinating, the conclusions are most powerful when the data can be distilled to direct observation of single release sites. Yet throughout, the data are presented as all-or-none discriminations at regions of interest (ROI) that may or may not represent single release sites. Specific criticisms are listed below.

1. One obvious challenge is distinguishing between a very small response and a non-responder. The data, throughout the paper, are categorized (yes/no) and presented as a percentage. Thus, the crux of the manuscript relies upon thoroughly demonstrating that the discrimination between a responder and non-responder is not confounded. For example, in several figures, nice example traces are provided showing a large initial response that results in a failure upon lowering extracellular calcium. But, beyond these examples, there are no quantitative data demonstrating that this is, indeed, routinely the case. Could the authors bin their data into quartiles based on the amplitude of the initial GCamp signal at 2.0 external calcium. Then, for each quartile, calculate the fractional silencing that occurs. If the fraction of silencing is constant across quartiles, then the authors have a good argument that the method of detection (GCamp) is not contributing to the observed effects.

2. Figure 6 uses GluSnFr to directly measure vesicle release, testing synapse silencing. Data are presented as the average of 15 single action potentials. Given that release is probabilistic, a silent synapse is a zero responder over some number of these trials, but we do not know how many. Two issues arise. First, individual successes could be averaged out to appear as a failure in the averaged trace. Second, a true release probability depends on the number of active zones that are present at a single R01. It seems that the true power of GluSnFr is not being deployed. Can the authors utilize spontaneous miniature amplitudes (or sucrose evoked miniatures) to achieve an estimate of single vesicle release event amplitude. Then, if sufficiently above the noise, define the release probability at high and low calcium.

3. In general, data from single RO1 would be nice to see, as opposed to averages and percentages per culture dish.

4. It would be important, in at least one experimental paradigm, to see that release returns to baseline following restoration of external calcium.

*Reviewer #2 (Recommendations for the authors):*

The authors illustrate two terminals in which the [Ca^2+^]e around 0.8 mM results in zero Ca^2+^ entry. They also show that at higher [Ca^2+^]e the terminal has detectable Ca^2+^ entry. In the following figure, they demonstrate that approximately 25% of the boutons are 'silent' even in 2 mM [Ca^2+^]e. What does this mean? Is there any evidence that they are functional boutons? This is inconsistent with the literature. In cortical pyramidal neuron axons, each AP results in detectable Ca^2+^ entry and the Ca^2+^ entry is 100% reliable from AP to AP. If there is no Ca^2+^ entry, it is always the consequence of the lack of an active zone (i.e. not functional bouton).

Keeping the superior properties of the synthetic dyes in mind, all [Ca^2+^] measurements should be repeated with high affinity synthetic dyes.

An interesting finding of the MS is that N and P/Q type Ca channels have no special role in Ca entry and in creating silent synapses. This should be emphasized.

*Reviewer #3 (Recommendations for the authors):*

1) Membrane excitability – Two observations are relevant here: The qualitative preservation of the phenomenon when two types of voltage gated calcium channels are blocked separately, and the large heterogeneity of the % of silenced boutons among neurons at a given extracellular calcium concentrations, which is at least as great as the range of modulation of the % of silenced synapses by extracellular calcium concentrations at single neurons. One then wonders if the findings might be attributed to a) the fidelity of the field potential-based stimulation system, that is, the degree to which neurons track the stimuli trains; b) the heterogeneity of neurons in this regard, c) this fidelity at different extracellular calcium concentrations for different neurons, and d) the identity of presynaptic sites analyzed in one run (are they all part of the same axon?). Along these lines, there is an assumption that the field potential-based stimulation system is the sole driver of excitation in these networks, which is reasonable given that excitatory synaptic transmission is mostly blocked pharmacologically (by CNQX and APV). Inhibitory transmission, however, was not blocked and thus, there is no guarantee that the inhibitory input neurons receive and its modulation by extracellular calcium does affect the degree to which neurons fire precisely and reliably at 20 Hz at all conditions. If it could be shown, at least for a substantial subset of the data, that all terminals analyzed for a particular neuron are part of an unambiguously identified axon stretch, with no branches (potential conduction failure points) and still demonstrate the claimed heterogeneity, this potential confound would be less of an issue.

2) It would seem that the x-axis intercept in Figure 1D (0.47 mM) predicts that at this concentration of extracellular calcium, pretty much all presynaptic terminals will be silent. Why not test this directly? This would nicely strengthen the findings.

3) Page 4, line 23. For the silenced population it is indeed expected, because, by definition, they did not respond and were consequently classified as silenced. As this is circular, the inclusion of these data in Figure 3D (and 6E) is a bit confusing and should probably be removed.

4) Figure 4 – end of legend – Data are fit as in D (should probably be E)

---

## [Author Response]

The essential revisions are representative of conversation among reviewers, attempting to synthesize the individual reviews.1. Two reviewers focused on a specific issue: The obvious challenge is distinguishing between a very small response and a non-responder. The data, throughout the paper, are categorized (yes/no) and presented as a percentage. Thus, the crux of the manuscript relies upon thoroughly demonstrating that the discrimination between a responder and non-responder is not confounded. Suggestions for how this might be achieved are presented, but these are not intended to be the only solution to addressing this issue – it is entirely up to the authors.

We thank the reviewers for raising this excellent point regarding the challenge of identifying nonresponding nerve terminals. To address this issue, we compared the frequency distributions of fluorescence responses (ΔF) across all nerve terminals separated into silent or responding populations (Figure 2 —figure supplement 1). This analysis reveals two key aspects: (1) the distribution shape of ΔF signals in the silent population is unchanged as [Ca^2+^]_e_ varies from 0.8 to 2.0 mM, while the distribution of responders shifts, as expected, to higher ΔF with increasing [Ca^2+^]_e_. This indicates that our cutoff criterion for defining silencing was not artificially being biased by the absolute size of the signal at these different concentrations. (2) We observe a marked, clear separation in these frequency distribution of the silent versus responding populations reinforcing that our cutoff criterium yields distinct populations. Last, our results for silencing at [Ca^2+^]_e_ 2 mM are consistent with previous publications using the same cutoff criterium or other definitions of presynaptic silencing, bolstering our observations and quantification in this manuscript. Conversely, for the first time to our knowledge, we have extended this methodology to investigate silencing at near-physiologic [Ca^2+^]_e_, discovering substantially different proportions of silent terminals within a physiologic range.

2. The data set could be broadly strengthened by including data from individual release sites, rather than simply presenting averages. This is particularly the case with the GluSnFr data where the probabilistic release of single vesicles could be resolved, inclusive of amplitudes associated with spontaneous release events (when possible).

We agree that extending our analysis to the level of single release sites would be interesting to establish whether they act independently within nerve terminals. With our approach, analysis of individual release sites is complicated by the possible activation of iGluSnFr by glutamate exocytosed from neighboring, non-transfected terminals and the variable number of release sites per terminal in CA1-CA3 hippocampal neurons. Thus, we would be limited in our ability to discriminate between neighboring release sites and characterize the proportion of silent release sites between different [Ca^2+^]_e_ concentrations. We have added this consideration as a limitation in the discussion of the manuscript (page 12, lines 7-15). However, the abolishment of Ca^2+^ influx at terminals despite a robust train of action potentials suggests that all release sites within a terminal are simultaneously silenced by this mechanism. Furthermore, the central importance of nerve terminals to neurotransmission, the ability to readily identify them through various targeted fluorescent proteins, and the widespread use of this marker in other studies of silencing underscore its biologic and experimental relevance.

3. Two reviewers highlight a circular argument regarding the classification of silent sites, that are then shown to be silent. This should be addressed either through changes to the figures or the text. Reviewer 3 (comment 3). Reviewer 2 (final comment).

As suggested by the reviewers, figures 3, 5, and 6 have been updated to eliminate the silent bars in these graphs.

4. It would be important, in at least one experimental paradigm, to see that release returns to baseline following restoration of external calcium.

We thank the reviewers for recommending inclusion of this important control experiment. We quantified the percentage of silent terminals and Ca^2+^ influx in neurons that were stimulated initially in [Ca^2+^]_e_ 2.0 mM, then 0.8 mM, and finally returned to 2.0 mM. We observed that both the proportion of silent terminals and the net influx of Ca^2+^ recovered to their initial values as [Ca^2+^]_e_ was reverted to its starting level (Figure 3 —figure supplement 2). Importantly, these results demonstrate that presynaptic silencing is a reversible process that is under the control of [a^2+^]_e_.

5. The issue of neuronal excitation can and should be evaluated. It is understood that these are sparse cultures. None-the-less, it remains possible that fields of view contain axons originating in more than one neuron, or in the case that they reside on different branches of the same axon, that branch point failures could 'cause' boutons to appear to function differently. A solution would be to systematically track whether sets of ROI unambiguously residing on single branches still function differently with respect to release (some being silenced and others not a lower calcium).'

We thank the reviewers for this suggestion to exclude the possibility that branch point failures are responsible for nerve terminal silencing. We examined this possibility with neurons expressing GCaMP6f in the cytosol to provide a good morphological fill of the axonal volume and co-expressing mRuby-synapsin to mark terminals. We performed the analysis for [Ca^2+^]_e_ 0.8 mM, the concentration we hypothesized would most likely cause branch point failure. To provide robust quantification, we limited our analysis to branches with at least 10 nerve terminals. This analysis demonstrates silent and responding terminals on the same branches of an axon, thereby excluding the possibility that these effects are driven by selective changes in excitability (Figure 2 —figure supplement 2). The overall percentage of silent terminals, though, is similar in this population of branches compared to the results of cell-wide percentages.

6. A core feature of the manuscript is the quantification of calcium entry via the genetically encoded calcium sensor. There was a healthy debate among reviewers regarding the necessity of utilizing synthetic dyes. On the one hand, the superior properties of the dyes allow for highly quantitative validation of existing transgenic reporters and central conclusions. On the other hand, this requires loading (likely through patch pipettes). Addressing the robustness of the existing measures, either through dyes or validation with additional variants of transgenic reporters would strengthen the work. Again, however, the precise nature of the experiment is entirely up to the authors.

We are grateful to the reviewers for highlighting this concern. Previous studies investigating presynaptic silencing (Kim and Ryan, 2013) utilized a less sensitive genetically-encoded calcium indicator, GCaMP3, and compared these measurements to the low-affinity, synthetic calcium dye, Magnesium Green. There was excellent agreement between the dye and GCaMP3, arguing against a limitation of GCaMP in detecting responses. Given the superiority of GCaMP6f to GCaMP3, we would expect a lower proportion of terminals classified as silent in our current experiments compared to the prior publication. However, we found a modest increase in the percentage of silent terminals (~15-20% compared to ~20-30%). We attribute the higher proportion of silent terminals in [Ca^2+^]_e_ 2.0 mM in the current studies to performing experiments at physiologic temperature as opposed to 30°C, which narrows the action potential waveform and thereby reduces Ca^2+^ influx. However, the overall similarity in our current results with this publication argues against a limitation of GCaMP as responsible for the silencing ofa^2+^ entry we observed.

7. Two reviewers, while agreeing that the GABA data are interesting, commented that the data are somewhat under-developed. On the one hand, the title emphasizes "neuromodulation" and this could be modified to better reflect the single experiment shown regarding GABA. At another level, reviewer #3 argues that "The interesting issue would be to reveal how the effect of GABAB activation depends on the [Ca^2+^]e. This information is essential to see whether there is indeed a shoulder in its effectiveness curve." The title likely should be changed, at a minimum.

We thank the reviewers for suggesting how to strengthen our conclusions of the impact of [Ca^2+^]_e_ on silencing mediated by the GABAB receptor, a GPCR that causes reduced Ca^2+^ entry via interaction of the Gβγ subunit with voltage-gated Ca^2+^ channels. As suggested, we investigated the effect of GABA_B_R agonism with baclofen on silencing at [Ca^2+^]_e_ 2.0 mM to compare to our previous results at 1.2 mM. Interestingly, the net change in percentage of silent terminals is substantially lower at 2.0 mM compared to 1.2 mM (7.6 ± 3.4% vs 20.5 ± 4.8%, respectively; Figure 7 —figure supplement 1). These results demonstrate that [Ca^2+^]_e_ substantially influences the synaptic response to GABA_B_R agonism, a GPCR that reduces Δ[Ca^2+^]_i_ through a canonical G-protein subunit pathway.

Additionally, throughout the text and in the title, we have replaced the term “neuromodulator” to “GPCR” or “GABA_B_R.”

8. There was discussion about the main conclusion of the manuscript. 'Thus, as with Ca^2+^ influx, SV recycling is modulated in an all-or-none manner by modest changes in [Ca^2+^]e around the physiological set point.' It seems equally likely that authors have identified a steeper continuum rather than 'all or none' control. As this is a central tenant of the manuscript, it could perhaps be addressed through changes in the text to shore up the argument, differently emphasized through data presentation or through the addition of new data.

Thank you for this insight. We have modified the text to point out that even if the Ca^2+^ influx is not completely silenced, the non-linear nature of the release process would effectively amplify the functional silencing effect (page 11, line 24 – page 12, line 6).

Reviewer #1 (Recommendations for the authors):While fascinating, the conclusions are most powerful when the data can be distilled to direct observation of single release sites. Yet throughout, the data are presented as all-or-none discriminations at regions of interest (ROI) that may or may not represent single release sites. Specific criticisms are listed below.1. One obvious challenge is distinguishing between a very small response and a non-responder. The data, throughout the paper, are categorized (yes/no) and presented as a percentage. Thus, the crux of the manuscript relies upon thoroughly demonstrating that the discrimination between a responder and non-responder is not confounded. For example, in several figures, nice example traces are provided showing a large initial response that results in a failure upon lowering extracellular calcium. But, beyond these examples, there are no quantitative data demonstrating that this is, indeed, routinely the case. Could the authors bin their data into quartiles based on the amplitude of the initial GCamp signal at 2.0 external calcium. Then, for each quartile, calculate the fractional silencing that occurs. If the fraction of silencing is constant across quartiles, then the authors have a good argument that the method of detection (GCamp) is not contributing to the observed effects.

Please see the response to #1 from the Essential revisions.

2. Figure 6 uses GluSnFr to directly measure vesicle release, testing synapse silencing. Data are presented as the average of 15 single action potentials. Given that release is probabilistic, a silent synapse is a zero responder over some number of these trials, but we do not know how many. Two issues arise. First, individual successes could be averaged out to appear as a failure in the averaged trace. Second, a true release probability depends on the number of active zones that are present at a single R01. It seems that the true power of GluSnFr is not being deployed. Can the authors utilize spontaneous miniature amplitudes (or sucrose evoked miniatures) to achieve an estimate of single vesicle release event amplitude. Then, if sufficiently above the noise, define the release probability at high and low calcium.

Please see the response to #2 from the Essential revisions.

3. In general, data from single RO1 would be nice to see, as opposed to averages and percentages per culture dish.

In addition to the inclusion of representative traces for the different biosensors used, we believe that the cumulative frequency distributions (Figure 2 —figure supplement 1, see also response to #1 from the Essential revisions) provide the best overall representation of the distribution of ΔF values from individual ROIs.

4. It would be important, in at least one experimental paradigm, to see that release returns to baseline following restoration of external calcium.

Please see the response to #4 from the Essential revisions.

Reviewer #2 (Recommendations for the authors):The authors illustrate two terminals in which the [Ca^2+^]e around 0.8 mM results in zero Ca^2+^ entry. They also show that at higher [Ca^2+^]e the terminal has detectable Ca^2+^ entry. In the following figure, they demonstrate that approximately 25% of the boutons are 'silent' even in 2 mM [Ca^2+^]e. What does this mean? Is there any evidence that they are functional boutons? This is inconsistent with the literature. In cortical pyramidal neuron axons, each AP results in detectable Ca^2+^ entry and the Ca^2+^ entry is 100% reliable from AP to AP. If there is no Ca^2+^ entry, it is always the consequence of the lack of an active zone (i.e. not functional bouton).

We thank the reviewer for this question. We presume the reviewer is referring to the article by Koester and Sakmann (2000), in which Ca^2+^ influx was measured in terminals following patching and loading of synthetic Ca^2+^ dyes. One important limitation of this study is that terminals were identified only morphologically as visible varicosities in the axon. This approach may have biased their measurements to particularly large nerve terminals that may not be representative. Moreover, a study performed by Frenguelli and Malinow (1996) in neocortical slices using a similar technique and post-hoc staining to verify that Ca^2+^ measurements were from boutons did observe failures of Ca^2+^ influx not attributable to failed AP propagation. Our results are consistent with these observations and previously published studies from the Ryan lab (Kim and Ryan, 2013).

Keeping the superior properties of the synthetic dyes in mind, all [Ca^2+^] measurements should be repeated with high affinity synthetic dyes.

Please see the response to #6 from the Essential revisions.

Reviewer #3 (Recommendations for the authors):1) Membrane excitability – Two observations are relevant here: The qualitative preservation of the phenomenon when two types of voltage gated calcium channels are blocked separately, and the large heterogeneity of the % of silenced boutons among neurons at a given extracellular calcium concentrations, which is at least as great as the range of modulation of the % of silenced synapses by extracellular calcium concentrations at single neurons. One then wonders if the findings might be attributed to a) the fidelity of the field potential-based stimulation system, that is, the degree to which neurons track the stimuli trains; b) the heterogeneity of neurons in this regard, c) this fidelity at different extracellular calcium concentrations for different neurons, and d) the identity of presynaptic sites analyzed in one run (are they all part of the same axon?). Along these lines, there is an assumption that the field potential-based stimulation system is the sole driver of excitation in these networks, which is reasonable given that excitatory synaptic transmission is mostly blocked pharmacologically (by CNQX and APV). Inhibitory transmission, however, was not blocked and thus, there is no guarantee that the inhibitory input neurons receive and its modulation by extracellular calcium does affect the degree to which neurons fire precisely and reliably at 20 Hz at all conditions. If it could be shown, at least for a substantial subset of the data, that all terminals analyzed for a particular neuron are part of an unambiguously identified axon stretch, with no branches (potential conduction failure points) and still demonstrate the claimed heterogeneity, this potential confound would be less of an issue.

Please see the response to #5 from the Essential revisions.

2) It would seem that the x-axis intercept in Figure 1D (0.47 mM) predicts that at this concentration of extracellular calcium, pretty much all presynaptic terminals will be silent. Why not test this directly? This would nicely strengthen the findings.

We thank the reviewer for this suggestion and we have conducted additional experiments comparing Δ[Ca^2+^]_i_ and the proportion of silencing in [Ca^2+^]_e_ 0.4 mM and 1.2 mM (Figure 1 —figure supplement 1 and Figure 3 —figure supplement 1). These experiments demonstrate that the overwhelming majority of terminals (93 ± 2%) are silent and the mean Δ[Ca^2+^]_i_ is nearly 0. Thus, these experiments support that the overwhelming majority of terminals have a non-zero threshold of [Ca^2+^]_e_ for [Δa^2+^]_i_.

3) Page 4, line 23. For the silenced population it is indeed expected, because, by definition, they did not respond and were consequently classified as silenced. As this is circular, the inclusion of these data in Figure 3D (and 6E) is a bit confusing and should probably be removed.

As suggested, we have removed the silent bars from these graphs.

4) Figure 4 – end of legend – Data are fit as in D (should probably be E)

Thank you for identifying this typographical error. It has been corrected.